# Fluid Balance and Carbohydrate Intake of Elite Female Soccer Players during Training and Competition

**DOI:** 10.3390/nu14153188

**Published:** 2022-08-03

**Authors:** Caroline A. Tarnowski, Ian Rollo, James M. Carter, Maria Antonia Lizarraga-Dallo, Mireia Porta Oliva, Tom Clifford, Lewis J. James, Rebecca K. Randell

**Affiliations:** 1Gatorade Sports Science Institute, PepsiCo Life Sciences, Global R&D, Leicestershire LE4 1ET, UK; ian.rollo@pepsico.com (I.R.); james.carter@pepsico.com (J.M.C.); rebecca.randell@pepsico.com (R.K.R.); 2School of Sport, Exercise and Health Sciences, Loughborough University, Leicestershire LE11 3TU, UK; t.clifford@lboro.ac.uk (T.C.); l.james@lboro.ac.uk (L.J.J.); 3FC Barcelona Medical Department, FC Barcelona, 08014 Barcelona, Spain; mlizarraga@ub.edu (M.A.L.-D.); mireia.porta@pl.fcbarcelona.cat (M.P.O.)

**Keywords:** hydration, soccer, female, sweat, fluid, carbohydrate intake, sodium

## Abstract

This study examined sweat rate, sweat sodium concentration [Na^+^], and ad-libitum carbohydrate and fluid intakes in elite female soccer players during training (*n* = 19) and a match (*n* = 8); eight completed both for comparisons. Body mass (kg) was obtained before and after exercise to calculate sweat rate. The sweat [Na^+^] was determined from absorbent patches on the thigh or back. Sweat rate, percentage body mass change, and sweat [Na^+^] for 19 players during training were 0.47 ± 0.19 L·h^−1^, +0.19 ± 0.65%, and 28 ± 10 mmol·L^−1^, respectively. Sweat rate was higher during a match (0.98 ± 0.34 L·h^−1^) versus training (0.49 ± 0.26 L·h^−1^, *p* = 0.007). Body mass losses were greater post-match (−1.12 ± 0.86%) than training (+0.29 ± 0.34%, *p* = 0.003). Sweat [Na^+^] was similar for training (29 ± 9 mmol·L^−1^) and a match (35 ± 9 mmol·L^−1^) (*p* = 0.215). There were no differences in match versus training carbohydrate intakes (2.0 ± 2.3 g·h^−1^, 0.9 ± 1.5 g·h^−1^, respectively, *p* = 0.219) or fluid intakes (0.71 ± 0.30 L·h^−1^, 0.53 ± 0.21 L·h^−1^, respectively, *p* = 0.114). In conclusion, female soccer players’ sweat rates were higher during a match than during training, and carbohydrate intakes were below recommendations for matches and training.

## 1. Introduction

Female soccer players cover ~10 km in distance during a match, using both the aerobic and anaerobic energy systems [1,2]. Such high-intensity intermittent running causes an elevation in core temperature [3], and the subsequent thermoregulatory response includes an increase in skin blood flow and the onset of sweating for evaporative heat loss [3]. High-intensity running also results in marked glycogen depletion in both muscle fiber types [4]. Elevations in core temperature, coupled with muscle glycogen depletion, can precipitate fatigue. In soccer, this may explain to some extent the reduction in distance covered and high-intensity running at the end of elite matches [5,6].

Male soccer players consume less fluid during matches [7] than during training [8], which may be due to the limited opportunities to rehydrate during a match. The high-intensity nature of soccer, combined with limited fluid availability, may put players at a higher risk of hypohydration. A simple method to assess the magnitude of hypohydration is to determine body mass changes pre- to post-exercise, with research suggesting that body mass losses of ≥2% may decrease technical and physical performance [9], while simultaneously increasing the risk of heat illness [10] and injury [11].

Few studies have investigated fluid balance, sweat rate, and sweat [Na^+^] during matches and training in elite female soccer players. In a study with international players, during two 90-min soccer-specific training sessions at 6 °C and 14 °C, hypohydration was similar and minimal (0.60% and 0.49%, respectively) [12]. Average sweat rates were also low (0.44 L·h^−1^ and 0.49 L·h^−1^, respectively) and no differences were found in sweat [Na^+^] [12]. Another study on female soccer players reported higher sweat rates (0.76 L·h^−1^) and mild hypohydration (1.2%) when training was completed in warmer conditions (30 °C) [13]. However, neither of these studies assessed fluid balance or sweat rate during match play. Simulated soccer match play (i.e., the Loughborough Intermittent Shuttle Test) in elite female players with fluid restriction (inducing ~2.2% hypohydration) elevated physiological (heart rate and blood lactate) and subjective (Rate of Perceived Exertion) responses compared to when fluid was consumed every 15 min [14]. To date, only one study has reported sweat rates and fluid balance during non-simulated match play, whereby elite female soccer players played two competitive matches in hot conditions (25 °C) [13]. The average sweat rate was 0.81 L·h^−1^, and hypohydration was 0.9% [13]. Based on our unpublished observations working in professional football, players often experience greater fluid losses during a match in comparison to training. However, to date, no study has investigated this. While previous studies report team averages for sweat loss and sweat [Na^+^] [12,13,14], large interindividual variations in sweat rate and sweat [Na^+^] have been reported in elite male soccer players when training [15]. However, the interindividual variation in hydration markers has yet to be reported for female players.

Furthermore, a recent review highlighted that there is currently limited research on the fluid and dietary intake of female soccer players [16]. In the studies that are available, carbohydrate intakes in female soccer players are low. For example, Wang et al., reported an average carbohydrate intake of ~11 g·h^−1^ during fourteen 90-min training sessions in collegiate soccer players [17], which is markedly lower than the 30–60 g·h^−1^ recommended for soccer players [18]. There is also evidence that carbohydrate intakes vary depending on the type of session; indeed, in elite male soccer players, carbohydrate intakes were lower during training (3 g·h^−1^) in comparison to a match (32 g·h^−1^) [19]. However, this comparison has not been made among elite female players. Sub-optimal intake of carbohydrates could impair soccer-specific performance [20]. In addition, consuming carbohydrates during training sessions that replicate match-play intensities can improve tolerance to carbohydrates during matches, potentially reducing any gastrointestinal complaints [21]. Knowledge of the carbohydrate intake of female players during training and matches could help in designing strategies to aid performance.

Therefore, the aim of the present study was to measure and compare sweat rate, sweat [Na^+^], carbohydrate consumption, and fluid intake in elite female soccer players during both a training session and a competitive match. We hypothesized that sweat rates, sweat [Na^+^], and carbohydrate intake would be greater during a match compared to training. However, fluid intake will be lower during a match due to limited access to fluid availability.

## 2. Materials and Methods

### 2.1. Study Participants

Nineteen elite professional female soccer players from Futbol Club (FC) Barcelona Femení team (Spanish first division; Primera División Femenina) participated in the present study [22]. Following explanation of the study details, all players completed written informed consent and a health screening questionnaire. The study was approved by the Research Ethics Committee of Loughborough University, U.K. (R16-P133). All players were accustomed to training and/or match durations of between 60 and 120 min, 3–6 times per week. The physical characteristics of the players were (mean ± SD): age: 25 ± 3 years; body mass: 61.0 ± 5.4 kg; stature: 169.4 ± 5.9 cm; body fat: 21.1 ± 4.1%; lean muscle mass: 45.1 ± 3.2 kg. Details regarding hormonal contraceptives or other medication use were not obtained.

### 2.2. Experimental Design

Measurements were made during one training session and one match during a competitive season, separated by two days in the month of November. The training duration was 85 min, which commenced at 10:15. The training consisted of typical field-based football exercises involving warm-up, active mobility, football drills and match play on a full-sized pitch. The match commenced at 16:00 and consisted of two 45-min halves with 15 min at half-time and 3 min additional time in each half, equating to a total time of 111 min. Both conditions were undertaken on an outdoor grass pitch.

Environmental conditions including ambient temperature (°C), relative humidity (%), wind speed (m·s^−1^) and wet bulb globe temperature (WBGT) were recorded (Kestrel 4500, Nielsen-Kellerman, Boothwyn, PA, USA) prior to both the training session and match, and at 15 min intervals until the cessation of exercise. Players wore training and match kits consisting of shorts, shirts, sports bras, socks, shin pads, and soccer boots. The training was coach-led, with no interference from the research team. Players did not have to adhere to any research-requested dietary controls.

### 2.3. Assessment of Fluid Balance

A pre-exercise urine sample was collected from all players 30 to 60 min prior to exercise. Samples were subsequently analyzed using a handheld refractometer (Atago 3730 Pen-Pro Dip-Style Digital Refractometer, Washington, USA) to provide an indication of pre-training hydration status. Hydration status was classified as follows: euhydrated (Urine Specific Gravity (USG) <1.020), minimally hypohydrated (USG 1.020–1.024) and hypohydrated (USG > 1.024) [23]. Thirst was assessed using a categorical thirst scale [24] whereby players self-selected a number corresponding to their current sensation of thirst with anchors at 1 representing ‘not thirsty at all’ and 7 indicating ‘very, very thirsty’.

Body mass (kg) was recorded prior to exercise in minimal clothing, which consisted of undergarments. Players were asked to void their bladders before being weighed. Body mass was measured again immediately following the training session and match. Players were provided with containers to collect any urine passed during the testing periods. Sweat rate and total sweat loss were calculated by assessing the change in body mass from pre- to post-exercise, correcting for total fluid and food intake and urine output during the testing period [25]. Relatively small changes in mass due to substrate oxidation and other sources of water loss were ignored [26].

### 2.4. Assessment of Sweat [Na^+^]

Samples of sweat were collected during both conditions using regional absorbent sweat patches (Tegaderm + Pad, 3M, Loughborough, UK). Patches were applied to the skin surface at two sites before exercise began; the right thigh and the right-hand side of the upper back (scapula). Prior to the patch application, the skin surface was cleaned with deionized water and dried using sterile gauze. The patches remained on the skin surface sites for the duration of exercise. Following exercise, the patches were removed and transferred immediately into separate 5 mL syringes using sterile tweezers. The syringe was used to aspirate sweat from the patch into a sterile cryovial, which was then sealed. All samples were analyzed within 24 h of cessation of exercise using an ion-selective electrode device (Horiba B-722) to measure sweat [Na^+^]. Samples from the thigh were used for analysis; however, if no sample could be obtained then the back sample was used. Regression equations can be used to predict whole-body [Na^+^] from regional sites [27]. Predicted whole body sweat [Na^+^] and sweat rate were used to determine sweat Na^+^ losses per hour of exercise.

### 2.5. Fluid and Carbohydrate Intake

During both conditions, players had ad libitum access to individually labeled water bottles and carbohydrate-electrolyte beverages (Gatorade Low-Calorie G2 (2%) and Gatorade (6%), PepsiCo, Ltd., Chicago, IL, USA). Players also had access to other carbohydrate sources such as bananas and sports gels. Players could access food and fluid at any point throughout the training session; however, during the match, players only had the opportunity to consume items at half time and during unscheduled breaks in play.

### 2.6. Statistical Analysis

Data are reported as mean and standard deviation (mean ± SD), with ranges reported in parentheses. Data were analyzed using Minitab software (version 17; Minitab Inc., State College, PA, USA). Data were considered normally distributed if Ryan-Joiner test was *p* > 0.05. Eight players completed both the full training session and match; therefore only data from these players were used for comparisons between the two occasions. All data were normally distributed therefore differences in variables between the training session and match were determined using paired samples t-tests. Statistical significance was accepted at *p* < 0.05. Relationships between variables were quantified using Pearson product moment correlation (r). The relationship between sweat rate and pre-exercise hydration status, sweat [Na^+^], fluid intake, and percentage change in body mass were calculated for both the training session and the match.

## 3. Results

The environmental conditions of the training session and the match are shown in Table 1. Ambient temperature, relative humidity %, wind speed, and WBGT were similar during both conditions (*p* > 0.05). The conditions in both the training session and the match were classified by WBGT as ‘cool’. The training session was rated as ‘moderate intensity’ by the coach.

### 3.1. Pre-Training Hydration Status

Prior to the training session, USG results indicated that nine players were euhydrated, seven players were minimally hypohydrated, and three players were hypohydrated. Prior to the match, USG results indicated that seven players were euhydrated and one player was minimally hypohydrated. For the eight players that completed both the training and match, USG was higher prior to the training session (1.020 ± 0.005) in comparison to the match (1.012 ± 0.007) (*p* = 0.020, Table 2). The thirst score was greater prior to the training session (5 ± 1, range; 3–5) compared to the match (2 ± 1, range; 1–3) (Table 2) (*p* = 0.024).

### 3.2. Fluid Balance

Three players produced urine during the training session and six during the match (Table 2). The average percent body mass change during training for all 19 players was +0.19 ± 0.65%, with 12 players gaining weight (range +0.1–0.7 kg). For the eight players that completed both the training session and the match, average body mass losses were significantly greater in the match condition (−1.12 ± 0.86%) compared to the training session (+0.29 ± 0.34%) (*p* = 0.003) (Table 2). Two players lost >2% of their pre-exercise body mass during the match.

### 3.3. Sweat Rate and Composition

Sweat rate and sweat [Na^+^] for all players (*n* = 19) that completed the training session can be found in Table 2. When comparing the two exercise conditions, the sweat rate was greater during the match than during the training session (0.85 ± 0.30 vs. 0.49 ± 0.26 L·h^−1^) (*p* = 0.016) (Table 2). When expressed as mmol·L^−1^, sweat [Na^+^] did not differ between the two conditions (*p* = 0.215); however, when expressed as mg·h^−1^ sweat [Na^+^] was significantly greater during the match (*p* = 0.037) (Table 2). Total Na^+^ losses were greater during the match (1302 ± 689 mg) compared to training (525 ± 381 mg) (*p* = 0.012) (Table 2). Large inter-individual variations in sweat rate (Figure 1) and sweat [Na^+^] (Figure 2) were evident.

During the training session (*n* = 19), a positive relationship was found between sweat rate and sweat [Na^+^] (r = 0.48, *p* = 0.037). No relationship was found between sweat rate and pre-training USG (r = −0.31, *p* = 0.194), percent change in body mass (r = −0.37, *p* = 0.121) or fluid intake (L·h^−1^) (r = 0.36, *p* = 0.127).

During the match (*n* = 8), no association was found between sweat rate and sweat [Na^+^] (r = 0.27, *p* = 0.511), pre-training USG (r = −0.44, *p* = 0.280) or fluid intake (L·h^−1^) (r = 0.33, *p* = 0.464). However, a negative relationship was found between sweat rate and percent change in body mass (r = −0.72, *p* = 0.042).

### 3.4. Fluid and Carbohydrate Consumption

Fluid (L·h^−1^) and carbohydrates (g·h^−1^) for all players (*n* = 19) during the training session can be found in Table 2. No differences were found in fluid and carbohydrate ingestion between the training session and the match (*p* = 0.114 and *p* = 0.219, respectively) (Table 3). Average fluid intake was 0.71 ± 0.30 L·h^−1^ during training and 0.62 ± 0.25 L·h^−1^ during the match (Table 3). Carbohydrate intake was 2.0 ± 2.3 g·h^−1^ throughout the training session and 0.9 ± 1.5 g·h^−1^ during the match (*p* = 0.219) (Table 3). The only source of carbohydrates consumed was carbohydrate–electrolyte beverages. In total, only five players consumed carbohydrates during the training session and three players during the match. Na^+^ consumption did not differ between the two conditions (Table 3).

## 4. Discussion

This is the first study to compare the sweat response, as well as carbohydrate and fluid intake of elite female soccer players during training and match conditions. The main findings were as follows: (1) sweat rate was significantly higher during the match compared to training; (2) there was large individual variation in both sweat rate and sweat [Na^+^]; (3) sweat [Na^+^] did not differ between training and the match; (4) carbohydrate intake was markedly below recommended intakes for both exercise occasions.

### 4.1. Sweat Rate

In the present study, the average sweat rate was significantly higher during the match compared to training. As environmental conditions and duration were similar during both, the difference is likely due to the higher exercise intensity of the match. Our results from the training session are consistent with previous studies, which report sweat rates of ~0.50 L·h^−1^ during training sessions of similar duration and environmental conditions [13]. However, the sweat rate during match play in the present study is higher than that reported in Australian national players (0.81 L·h^−1^), despite the Australian match being played in warmer conditions (25 °C) [13]. Speculatively, this could be due to a difference in intensity, as previous studies have found high-speed running to be greater in the domestic Spanish soccer league [28] compared to the Australian league [29].

We also observed a large variation in sweat rates during the match and training session. This data is consistent with studies in male soccer players, where large individual variation was found in sweat rates during both training and matches [15,30]. Therefore, players should be tested during a range of environmental conditions and intensities of exercise in order to make personalized recommendations [18]. Finally, it is recognized that competition may add additional anxiety and/or stress to the player, which may influence the sweating response [31]. However, it was beyond the scope of the present study to measure this.

### 4.2. Sweat Sodium

We found no differences in sweat [Na^+^] between the training session and the match. Only one other study has reported the sweat [Na^+^] of elite female soccer players. Kilding et al. [12] reported an average sweat [Na^+^] of 28 mmol·L^−1^ during two training sessions. In the present study, when the sweat [Na^+^] was combined for training and match-play, a similar mean concentration was observed (30 mmol·L^−1^). However, unlike in the present study, Kilding and colleagues did not report individual variation. We found large individual variations in sweat [Na^+^] during the training session and match (Figure 2). However, sweat [Na^+^] did not differ for individuals between training and the match. This may suggest that in these conditions, sweat [Na^+^] data from training could be extrapolated to inform match day hydration strategies. In a recent laboratory study, sweat [Na^+^] significantly increased when samples were taken from several upper body regional sites in response to increased exercise intensity, but there were no changes to sweat [Na^+^] in the lower body (thigh and calf) [32]. However, this is difficult to confirm in the present study without data from additional anatomical regions. Future field research measuring sweat [Na^+^] from several upper and lower body sites is needed.

### 4.3. Fluid Intake

Fluid intakes were not significantly different between the training session and match. Despite higher sweat rates during the match, fluid intake during the training session was expected to be higher, as there was an increased opportunity to consume fluid compared to a match, where drinking is restricted to certain time periods [33]. The average total fluid intake during training in the present study (1.01 L) was more than double that recorded by Kilding et al. [12] (0.40 L) when training for a similar duration. The reason for the discrepancy between fluid intakes is unclear; perhaps fluid was more accessible in the present study, or there are inherent differences in cultural or nutritional practices between the players in the two studies.

### 4.4. Pre-Exercise Hydration Status

Whilst the limitations of USG as a hydration assessment are well described [34], five out of the six players commenced training with a USG > 1.020, possibly indicative of hypohydration; whereas prior to the match, only one player had a USG > 1.020. In addition, thirst scores were greater prior to training compared to the match. This may be due to the later start time of the match compared to training, allowing players more time to consume fluids and hydrate. A study in elite female soccer players found that prior to matches and training ~47% of players were severely hypohydrated (USG > 1.030) and ~33% of players were significantly hypohydrated (USG > 1.020) [35], suggesting many players are not adequately re-hydrating between matches and training. Our data also demonstrates that the extent of hypohydration during exercise may be underestimated given some players are already in a hypohydrated state. Future research should be encouraged to record true baseline measures of body mass as well as other valid markers of hydration to more accurately record hydration status [34]. In parallel, practitioners would be advised to work closer with their players on pre-exercise hydration strategies and routines to ensure euhydration prior to training and competition.

### 4.5. Carbohydrate Ingestion

Carbohydrate ingestion is recommended to support team sports performance [36]. For high-intensity intermittent exercise lasting >60 min, the recommended intake is 30–60 g·h^−1^ [18]. In the present study, female soccer players ingested an average of 2.0 g·h^−1^ during the training session and 0.9 g·h^−1^ during the match, with players opting for the 2% carbohydrate beverage (as opposed to the 6% carbohydrate beverage). The players had more opportunities to consume carbohydrates during training, so the fact that the intakes were low during the match and training suggests that this is due to preference rather than access or opportunity to ingest carbohydrates. Not meeting carbohydrate intake recommendations during training and matches could impair soccer-specific performance [37,38,39]. In addition, not meeting carbohydrate recommendations throughout the day could put players at risk of low energy availability; indeed, a recent study by Moss et al., showed that 23% of top league players had low energy availability during the in-season period, partly due to sub-optimal carbohydrate ingestion [40]. Future research should explore why female players self-selected lower-carbohydrate options (i.e., 2% solution more than a 6% solution) and did not adjust carbohydrate intake in response to the change in exercise demands. A better understanding of female soccer players’ knowledge of carbohydrates could help inform tailored educational sessions [41], which, together with hydration education, should be an additional focus of sports nutrition education for practitioners working in elite female soccer.

### 4.6. Limitations, Strengths and Future Research

The present study was descriptive and, thus, several limitations are acknowledged. Firstly, we did not collect any measures of exercise volume or internal physiological responses, and therefore we were unable to compare the exercise intensity during the two exercise sessions. Secondly, potassium concentrations were not analyzed in sweat, which is useful for quality control for sweat samples [42]. Despite this, best practices for sweat collection (clean skin, avoided patch saturation, gloves, etc.) described by Baker [42] were followed, so that we have confidence in our analysis. Other factors reported to influence sweat [Na^+^] such as hormones [43] and diet [44] were not recorded or controlled for prior to testing, due to the nature of field/descriptive studies. A recent study found no difference in the degree of hypohydration, net fluid balance, electrolyte balance, and thirst intensity in the mid-luteal phase compared to the late follicular phase in female athletes; therefore it is unlikely that the phase of the menstrual cycle would have impacted the results of the present study [45]. However, it is imperative that future research follows methodological guidelines pertaining to high-quality research in female athletes [46]. Another limitation of the present study was that fluid intake was only assessed during exercise. The pre-training spot USG measure did not allow for a complete understanding of the players’ hydration status before training. Assessing first-morning USG, 24-h urine volumes, and/or 24-h fluid intake prior to training would provide insights into the influence of baseline hydration on sweat rates and sweat [Na^+^].

## 5. Conclusions

In conclusion, the sweat rates of elite female soccer players are higher during matches compared to training. The large interindividual variation in sweat rate and sweat [Na^+^] during training and matches supports individualized hydration recommendations for this cohort. The low carbohydrate intake in this population during both training and matches warrants further investigation and suggests greater intervention is required from sports nutritionists within elite female soccer.

## Figures and Tables

**Figure 1 nutrients-14-03188-f001:**
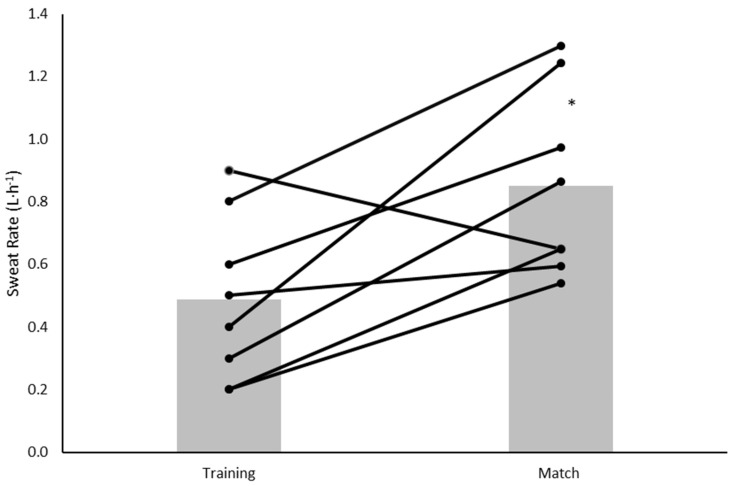
Individual player and mean sweat rates (L·h^−1^) during the training session and match. Black lines = individual data, grey bars = mean. * indicates training significantly different to training (*p* < 0.05).

**Figure 2 nutrients-14-03188-f002:**
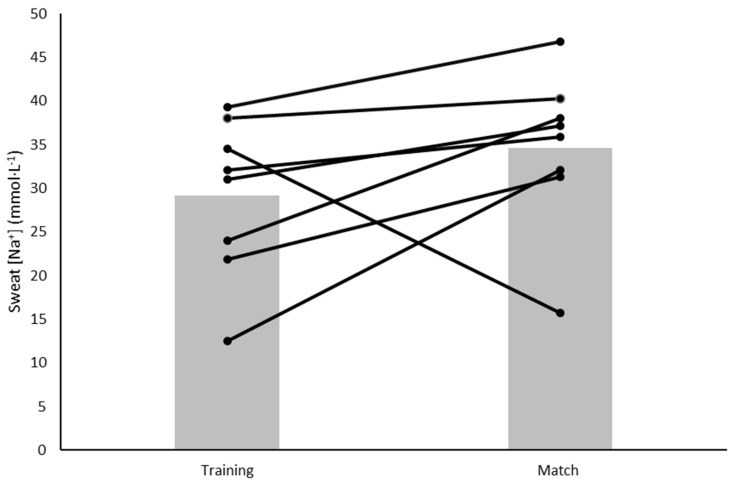
Individual and mean sweat [Na^+^] (mmol·L^−1^) during the training session and match. Black lines = individual data, grey bars = mean.

**Table 1 nutrients-14-03188-t001:** Environmental conditions for the training and match (mean ± SD).

	Duration (min)	Temperature (°C)	Relative Humidity (%)	Wind Speed (m·s^−1^)	WBGT (°C)
**Training**	85	18.6 ± 1.3	57 ± 5	0.0 ± 0.0	13.5 ± 1.2
**Match**	111	18.8 ± 0.7	66 ± 1	0.1 ± 0.1	14.8 ± 0.6

**Table 2 nutrients-14-03188-t002:** Sweat test variables recorded before and during training and match (mean ± SD (range)). Training (*n* = 8) and match (*n* = 8) only includes data of players that completed both the full training session and match. * indicates training significantly different to match (*p* < 0.05).

	Training (*n* = 19)	Training (*n* = 8)	Match (*n* = 8)
**Pre-training USG**	1.020 ± 0.005 (1.009–1.029)	1.020 ± 0.005 (1.010–1.025) *	1.012 ± 0.007 (1.004–1.023)
**Pre-Body Mass (kg)**	60.4 ± 5.6 (45.0–68.1)	62.1 ± 3.8 (57.4–68.1)	62.3 ± 3.9 (57.9–68.1)
**Post-Body Mass (kg)**	60.5 ± 5.8 (44.9–68.4)	62.3 ± 3.9 (57.9–68.4) *	61.6 ± 4.2 (57.1–67.8)
**Δ Body Mass (kg)**	+0.1 ± 0.4 (−0.7–0.7)	+0.2 ± 0.3 (−0.5–0.5) *	−0.7 ± 0.5 (−1.5–0.1)
**Percent Body Mass Change (%)**	+0.19 ± 0.65 (−1.27–1.30)	+0.29 ± 0.57 (−0.93–0.87) *	−1.12 ± 0.86 (−2.34–0.24)
**Urine Output (mL)**	87 ± 185 (0–737)	104 ± 257 (0–737)	107 ± 83 (0–241)
**Sweat Rate (L·h^−1^)**	0.47 ± 0.19 (0.20–0.90)	0.49 ± 0.26 (0.20–0.90) *	0.85 ± 0.30 (0.54–1.30)
**Total Sweat Loss (L)**	0.68 ± 0.28 (0.20–1.30)	0.70 ± 0.38 (0.20–1.30) *	1.58 ± 0.55 (1.00–2.40)
**Sweat Na^+^ Losses (mg·h^−1^)**	323 ± 208 (55–778)	370 ± 269 (55–778) *	704 ± 373 (214–1370)
**Sweat [Na^+^] (mmol·L^−1^)**	28 ± 10 (12–41)	29 ± 9 (13–39)	35 ± 9 (16–47)
**Pre-exercise Thirst Score**	3 ± 1 (1–5)	4 ± 1 (3–5) *	2 ± 1 (1–3)

**Table 3 nutrients-14-03188-t003:** Sweat test variables recorded before and during training and match (Mean ± SD (range)). Training (*n* = 8) and match (*n* = 8) only includes data of players that completed both the full training session and match. * indicates training significantly different to match (*p* < 0.05).

	Training (*n* = 19)	Training (*n* = 8)	Match (*n* = 8)
**Fluid Intake (L·h^−1^)**	0.62 ± 0.26 (0.15–1.07)	0.71 ± 0.30 (0.15–1.07)	0.53 ± 0.21 (0.24–0.81)
**Total Fluid Intake (L)**	0.88 ± 0.36 (0.22–1.51)	1.01 ± 0.42 (0.22–1.51)	0.99 ± 0.39 (0.44–1.49)
**Carbohydrate Intake (g·h^−1^)**	1.6 ± 2.5 (0.0–9.0)	2.0 ± 2.3 (0.0–6.0)	0.5 ± 0.2 (0.0–2.2)
**Na^+^ Consumption (mg·h^−1^)**	74 ± 117 (0–426)	87 ± 105 (0–276)	35 ± 53 (0–142)
**Total Na^+^ Consumption (mg)**	105 ± 166 (0–604)	123 ± 149 (0–391)	65 ± 98 (0–262)

## Data Availability

The data presented in this study are available on request from the corresponding author and the permission of all parties involved in the study. The data are not publicly available due to privacy.

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
