# Peer review of "Fluid Balance and Carbohydrate Intake of Elite Female Soccer Players during Training and Competition"

_nutrients, 2022, doi:10.3390/nu14153188_

Round 1
Reviewer 1 Report
No rationale stated for hypothesis that sweat rates, sweat and CHO intake will be greater during a match compared to training. Is this hypothesis based on the data from males? This should be determined by the intensity and period of training and individual training status of hydration among females.
What types of exercise during training were performed, aerobic, anaerobic, or combined? Training session was not clearly stated in the manuscript. Did every soccer players have same training(intensity, duration and so on) based on their position on the field? Also, did they run same amount of distance during the match? Without these data, the results of the study cannot be fully explained.
In the results, table 2 and 3, why the authors divided into three groups including training(n=19), training(n=8), and match(n=8)? No explanation for this was stated in methods section. In figure 1 and 2 what if remove one person who showed opposite way from others?
More data are likely to be needed to discuss the results of this study's measures. For example, values such as sweat rate and carbohydrate intake may vary depending on individual position, training time vs. match time, and training level, especially in the case of soccer.
Reviewer 2 Report
In this manuscript, the authors describe "Fluid balance and Carbohydrate Intake of Elite Female Soccer Players During Training and Competition". Although this manuscript is quite interesting, it would be helpful if the authors address the following concerns:
1) In line 57, Na+ is not consistent with others throughout the manuscript. It should be in square brackets...ie [Na+].
2) In line 98, the partial sentence..." prior to the both the training session..." the words in bold should be removed as the present sentence does not make sense.
3) Although the study was centered on female elite athletes, the authors did not define nor provided a criteria regarding what constitute an athlete to be considered an elite. It would be helpful to readers if the authors explain in clear terms what parameters they used.
4) Did the study took into consideration the kind of medications these athletes were on (if any), and the potential impact it may have on their sweating rates during training and in competitive soccer matches? It would be helpful if the authors clarify these.
5)In line 246 - 250, because sweat rates will be different from different anatomical sites of the body, which will also be different from different individuals, how does the present study explains this?
6) In lines 221 - 222 and lines 228 - 230, Apart from running this study in different environmental temperature conditions, and while the present study shows a significantly higher sweating rates during soccer matches compared to training, it would be helpful to readers if the authors also consider other ancillary factors such as the players stress levels, nervousness and anxiety (during competitive soccer matches) and their impact on sweating rates. This will help provide the big picture regarding what is going on.
In short, this manuscript will benefit its targeted audience if the above concerns are addressed.
Round 2
Reviewer 1 Report
I do not have any more review comments.
Reviewer 2 Report
In this manuscript the authors describe "Fluid Balance and Carbohydrate Intake of Elite Female Soccer Players During Training and Competition". While some concerns were raised in certain aspects of the study (in the initial review), the authors were able to adequately addressed all those concerns raised, and made the needed updates/additions as well as supporting them with key references where appropriate.
This study therefore has the propensity to benefit a wider audience as written.